# Differential Exoproteome and Biochemical Characterisation of *Neoparamoeba perurans*

**DOI:** 10.3390/microorganisms9061258

**Published:** 2021-06-09

**Authors:** Kerrie Ní Dhufaigh, Natasha Botwright, Eugene Dillon, Ian O’Connor, Eugene MacCarthy, Orla Slattery

**Affiliations:** 1Marine and Freshwater Research Centre, Galway-Mayo Institute of Technology, Co. Galway, H91 T8NW Eircode, Ireland; ian.oconnor@gmit.ie (I.O.); eugene.mccarthy@gmit.ie (E.M.); 2CSIRO Agriculture and Food, Livestock & Aquaculture, Queensland Biosciences Precinct, 306 Carmody Road, Brisbane, QLD 4067, Australia; natasha.botwright@csiro.au; 3Conway Institute of Biomolecular & Biomedical Research, University College Dublin, Co. Dublin, D04 V1W8 Eircode, Ireland; eugene.dillon@ucd.ie; 4Department of Biopharmaceutical and Medical Science, Galway-Mayo Institute of Technology, Co. Galway, H91 T8NW Eircode, Ireland; orla.slattery@gmit.ie

**Keywords:** amoebic gill disease, exoproteome, LC-MS/MS, virulence factors, protease activity, cytotoxicity, proteomics

## Abstract

Infection with the protozoan ectoparasite *Neoparamoeba perurans*, the causative agent of AGD, remains a global threat to salmonid farming. This study aimed to analyse the exoproteome of both an attenuated and virulent *N. perurans* isolate using proteomics and cytotoxicity testing. A disproportionate presence of proteins from the co-cultured microbiota of *N. perurans* was revealed on searching an amalgamated database of bacterial, *N. perurans* and Amoebozoa proteins. LC-MS/MS identified 33 differentially expressed proteins, the majority of which were upregulated in the attenuated exoproteome. Proteins of putative interest found in both exoproteomes were maltoporin, ferrichrome-iron receptor, and putative ferric enterobactin receptor. Protease activity remained significantly elevated in the attenuated exoproteome compared with the virulent exoproteome. Similarly, the attenuated exoproteome had a significantly higher cytotoxic effect on rainbow trout gill cell line (RTgill W1) cells compared with the virulent exoproteome. The presence of a phosphatase and serine protease in the virulent exoproteome may facilitate AGD infection but do not appear to be key players in causing cytotoxicity. Altogether, this study reveals prolonged culture of *N. perurans* affects the exoproteome composition in favour of nutritional acquisition, and that the current culturing protocol for virulent *N. perurans* does not facilitate the secretion of virulence factors.

## 1. Introduction

*Neoparamoeba perurans* (synonym *Paramoeba peruans* [1]) is the marine ectoparasitic agent of amoebic gill disease (AGD) in finfish aquaculture [2]. Feehan et al. [1] demonstrated that *Neoparamoeba* and *Paramoeba* are synonymous names by using nuclear SSU rDNA to declare them phylogenetically inseparable. Previously reported in Tasmania [3], AGD’s prevalence and geographic range has expanded to all marine *Salmo salar* L. farming countries including Norway [4], Chile [5], Scotland [6], and Ireland [7]. High morbidity and mortality of affected finfish and economic losses from expensive, curative treatments has led to AGD becoming a significant health problem for aquaculture [6,8]. Infection presents as raised white mucoid lesions on the gill surface [9]. Histopathology shows epithelial hyperplasia on the host’s primary and secondary lamellae [10,11].

The secretome or exoproteome of pathogenic organisms has gained increased attention in recent years, as many secreted proteins are known to be involved in host-pathogen associations, such as: adhesion, morphogenesis, and toxicity to the host’s immune system [12,13]. Secreted proteins can be regarded as virulence factors of infectious diseases as they play a role in disease progression [14]. It is well documented that protozoan parasites are known to secrete glycoproteins, proteinases such as elastases, metalloproteases, as well as serine and cysteine proteases that facilitate host tissue invasion [15] and augment tissue destruction [16]. Furthermore, pathogenic amoebae have the ability to produce hydrolytic enzymes that cause detrimental damage to a host’s membrane [17].

It has been hypothesised that *N. perurans* secretes an extracellular product during culture that has virulence implications for AGD development and host-parasite interactions [18,19]. The authors hypothesising such an extracellular product, suggest a cytolytic enzyme was responsible for the observed cytopathic effects in the fish cell line, CHSE-214. Furthermore, Bridle et al. [19] demonstrated the inability of an avirulent isolate of *N. perurans* (>3 years in culture), to establish the cytopathic effect that was observed in the virulent isolate. Despite Bridle et al. [19] demonstrating a cytopathic effect from a virulent culture, the causative proteins responsible for the observed host cytopathic response were not identified. Therefore, this study aims to describe the complete extracellular products of a previously confirmed attenuated (4-year-old culture) and virulent (70 day old) *N. perurans* culture [20], using a gel-free LC-MS/MS approach.

This study employed protein identification, protease activity and inhibition assays to confirm the presence and class of proteases in the extracellular proteome of an attenuated and virulent *N. perurans* culture. The cytotoxic effect of each exoproteome on the gill cell line, RTgill W1, was evaluated using the Alamar Blue assay [21]. Proteomic approaches such as liquid-chromatography tandem mass spectrometry (LC-MS/MS) represent a high throughput technique that can map the extracellular proteome of significantly important parasites [15,22,23], thus elucidating their pathogenicity.

## 2. Materials and Methods

### 2.1. The Neoparamoeba perurans Culture

*N. perurans* trophozoites were isolated from naïve Atlantic salmon gills affected by AGD located on a commercial farm in the west of Ireland. The naïve smolts were issued an AGD gill score [24] of 1. An attenuated culture of *N. perurans*, previously shown to have lost virulence [20] was used as the attenuated isolate (4 years in culture at the time of protein extraction). To acquire a virulent isolate, fresh trophozoites were collected by swabbing AGD affected gills and placing swabs in 0.2 µm filtered sterile seawater for 4 h to release trophozoites from the swab into the seawater. This combined mixture was subsequently plated and maintained as xenic cultures at 16 °C on marine malt yeast agar plates (MYA; 0.01% malt, 0.01% yeast, 2% Bacto Agar, sterile sea water at 30% ppt) overlaid with 7 mL of 0.2 µm filtered sea water [25]. The virulent isolate was maintained for 70 days before protein extraction. The virulence of the virulent isolate was proven in a separate challenge trial to this study (data not shown).

Inoculated plates were washed weekly with 7 mL of sterile seawater to control bacterial growth. Amoebae exhibited characteristic pseudopodia and were sub-cultured weekly by transferring free-floating cells to fresh MYA plates. Confirmation of *N. perurans* identity was performed using real time PCR, as previously described by Downes et al. [26].

### 2.2. Extraction of Extracellular Proteome

Both the attenuated and virulent cultures were harvested upon reaching 70% confluency (approximately two weeks of culturing). Trophozoites were mechanically scraped from several MYA plates and pooled into 50 mL falcon tubes. The trophozoites were centrifuged at 1000× *g* for 10 min at 4 °C. The supernatant of the cultures was aspirated into fresh 50 mL falcon tubes followed by centrifugation at 10,000× *g* for 10 min at 4 °C to further remove bacterial and amoebae debris. After centrifugation, the supernatant was passed through a 0.45 µm filter (Pall Corporation, Port Washington, NY, USA) and subsequently a 0.25 µm filter (Pall Corporation) to obtain a cell free filtrate containing the putative extracellular proteome or exoproteome. The filtrate was concentrated 10-fold using Amicon centrifugal filters with a 3 kDa molecular-weight cut off (Millipore Corp, Watford, UK). Protein quantification was measured with the BCA assay (Pierce™Thermofisher Scientific, Waltham, MA, USA).

### 2.3. In-Solution Trypsin Digestion

Concentrated exoproteomes of the attenuated and virulent cultures were adjusted to 100 µg/mL of protein. Buffer was exchanged using centrifugal concentrators in 20 mM Tris-HCl pH 7.5 for desalting purposes. Sample preparation occurred using the PreOmics kit (Munich, Germany) according to the Pellet and Precipitated Protein iST protocol with modifications as described by Kulak et al. [27]. Briefly, desalted exoproteomes were concentrated to 100 µL and 300 µL of the lysis buffer was added. The sample was concentrated to a final volume of 50 µL and heated at 95 °C for 5 min on a heating block. Samples were then placed in iST cartridges for trypsin digestion (Trypsin/LysC) at 37 °C for 2.5 h with mixing. Peptides were acidified and subsequently washed using the wash 1 and wash 2 buffers supplied. Peptides were eluted in 120 µL and dried using a vacuum centrifuge (Savant Speedvac, Thermo Fisher Scientific, San Jose, CA, USA). After drying, peptides were resuspended in 40 µL LC load buffer.

### 2.4. Liquid Chromatography Tandem Mass Spectrometry

Peptide fractions were analysed on a quadrupole Orbitrap (Q-Exactive, Thermo Scientific, San Jose, CA, USA)) mass spectrometer equipped with a reversed-phase NanoLC UltiMate 3000 HPLC system (Dionex LC Packings, now Thermo Scientific, San Jose, CA, USA). Samples were separated on an in-house made 10 cm reversed phase capillary emitter column (inner diameter 75 μm, 3 μm ReproSil-Pur C18-AQ media (Dr. Maisch GmbH, Ammerbuch-Entringen, Germany). Peptide samples were eluted with a linear gradient from 1–27% buffer B containing 0.5% acetic acid (AA) 97.5% acetonitrile (ACN) in 58 min at a flow rate of 250 nL/min. The injection volume was 5 μL.

### 2.5. Raw Data Processing and Label Free Quantification (LFQ)

Raw data from the Orbitrap Q-Exactive were processed using MaxQuant version 1.6.14.0 for identification of proteins [28], incorporating the Andromeda search engine and MaxQuant’s contaminants fasta file [29]. To identify peptides and proteins, MS/MS spectra were matched to a custom database of proteins from *N*. *perurans* (20,887 proteins (v2, 07/08/2019, CSIRO)), combined with a UniProt reference proteome database of Amoebozoa proteins (109,415 proteins), as well as proteins from a *Paramoeba* UniProtKB taxonomy search (5001 proteins). The inclusion of proteins from a broad range of amoebae in the Amoebozoa dataset was to both facilitate and validate the identification of amoeba-specific proteins. Co-culturing bacteria of *N. perurans* contributed to the protein identifications in the cytoplasmic protein investigation of *N. perurans* [20], therefore, it is imperative to distinguish between *N. perurans* specific and bacterial specific proteins. The microbial communities of the attenuated and virulent cultures were previously evaluated using 16S rRNA gene sequencing [20]. The key genera identified as part of this analysis were used to inform the creation of a bacteria protein database consisting of 148,582 proteins. This bacteria database was used in conjunction with the aforementioned amoebae protein databases. All proteins, except for those from the *N. perurans* database, were downloaded on 8 May 2020 from UniProt. Details of the total protein count from each species included in the database can be found in Appendix A.

All searches were performed with tryptic specificity allowing two missed cleavages. The database searches were performed with carbamidomethyl (C) as fixed modification and acetylation (protein N terminus) and oxidation (M) as variable modifications. Match between runs was allowed and proteins were quantified using the LFQ algorithm in MaxQuant [30]. Mass spectra were searched using the default setting of MaxQuant namely, a false discovery rate of 1% on the peptide and protein level. For bioinformatic analysis, LFQ values obtained in MaxQuant were imported into Perseus (v. 1.6.10.50) [31] software. Reverse hits, contaminants and peptides identified by site were filtered out. LFQ values were log2 transformed and differentially expressed proteins between samples were identified using a two-way Students *t*-test followed by Benjamin-Hochberg false discovery rate (FDR) correction. Hierarchical clustering was generated for the statistically significant proteins using Z-score protein intensities for the proteins with *p* < 0.05.

### 2.6. Enzymatic Characterization

Protease activity of the attenuated and virulent concentrated exoproteomes was evaluated as previously described [32,33] with modifications. Briefly, 250 µL of 2% azocasein dye (Sigma-Aldrich, St. Louis, MO, USA) was added to 100 µL of sodium phosphate buffer pH 8, serving as the reaction buffer. A 100 µL aliquot of exoproteome (40 µg/mL), was added to the reaction buffer for both the attenuated and virulent exoproteomes. Aliquots of 100 µL of 2 mg/mL Trypsin (Sigma-Aldrich) served as the positive control and 100 µL sterile seawater (0.45 µm filtered and autoclaved) for the negative control. Samples were incubated overnight at room temperature (20–22 °C) on a rocker. Reactions were stopped by the addition of 250 µL of 10% trichloroacetic acid (TCA) and centrifuged at 6000× *g* for 5 min at room temperature. A 100 µL aliquot of the sample supernatants was added to 100 µL of 1 N sodium hydroxide (NaOH) in a 96 well plate. Optical density was recorded at 450 nm on a Multiskan Sky microplate reader (ThermoFisher, Waltham, MA, USA) using SkanIt software (ThermoFisher, Waltham, MA, USA). The absorbance of the sterile seawater sample was subtracted from all other assay absorbance values. Protease activity was presented as a percentage of the positive control.

### 2.7. Enzyme Inhibition Assay

The protease assay as described in Section 2.6 was modified to include the presence of various protease inhibitor classes to determine protease families found in the *N. perurans* exoproteome. Two protease inhibitors were selected, phenylmethylsulfonyl fluoride (PMSF) targeting the serine protease family and ethylenediaminetetraacetic acid (EDTA) targeting metalloproteinases. Exoproteome samples, trypsin, and seawater controls were preincubated with 1 mM PMSF or 1 mM EDTA at room temperature for 2 min prior to the addition of azocasein dye and phosphate buffer. The experiment followed procedures as described in Section 2.6.

### 2.8. RTgill W1 Culture and Assay Preparation

The *Oncorhynchus mykiss*, (rainbow trout) gill cell line RTgill W1 was purchased from the American Tissue Culture Type Collection (CRL 2523) and cultivated according to Bols et al. [34] and Dayeh et al. [21]. RTgill W1 cells were cultivated at 19 °C on tissue culture treated T-75 cm^2^ flasks (VWR Collection) in Leibovitz’s L-15 media (Hyclone Laboratories, Inc., Logan, UT, USA) supplemented with 10% foetal bovine serum (FBS) (Gibco, Thermofisher Scientific, Waltham, MA, USA) and an antibiotic/antimycotic solution (10,000 U/mL Penicillin G, 10,000 µg/mL Streptomycin, 25 µg/mL Amphotericin B (Hyclone Laboratories, Inc., Logan, UT, USA). RTgill W1 cells were cultivated in 100% atmospheric O₂, Forma Series II (Thermo Scientific, San Jose, CA, USA).

Once 70% confluency was achieved, RTgill W1 cells were prepared for the cytopathic assays. Briefly, cells were detached by trypsin/EDTA solution (Biowest, Nuaillé, France), resuspended in 5 mL of L-15 containing FBS and cell number counted by haemocytometer (Neubauer, Hawksley, Lancing, United Kingdom ). Cells were seeded overnight at 19 °C at a cell density of 25,000 cells/well in a fluorescent 96-well plate (Corning Incorporated, Corning, NY, USA). After 24 h, media was removed from the wells and test media was added. Exoproteomes from *N. perurans* attenuated and virulent cultures were buffer exchanged into phenol red-free L-15 media using centrifugal concentrators. A total of 40 µg of protein from each exoproteome sample was added to test wells in a final volume of 200 µL. Positive controls were treated with phenol red-free L-15 media only, while 0.5% Triton X-100 (Sigma-Aldrich, St. Louis, MO, USA) served as the negative control. Media only was added to no-cell control wells to monitor background fluorescence. The final volume in each well was 200 µL. After 24 h of test exposure, RTgill W1 viability was assessed.

### 2.9. Cytotoxicity Alamar Blue Preparation

Alamar blue was prepared as per Dayeh et al. [21]. A 5% (*v*/*v*) alamar blue solution was prepared in sterile filtered PBS pH 7.4, prior to starting the assay. This assay was performed three times independently with the exoproteomes of the attenuated and virulent cultures. Media was removed from the 96-well plate and 200 µL of alamar blue was pipetted into each well. The plate was incubated in the dark for 30 min at 19 °C. Fluorescence measurements were carried out using a Fluoroskan™Microplate Fluorometer (Thermofisher, Waltham, MA, USA) at 530 nm (excitation) and 590 nm (emission).

## 3. Results

### 3.1. Exoproteome Identification Using Differential Proteomics

Results of the LC-MS/MS showed that 38 proteins were identified from a combination of bacterial and amoebae (Amoebozoa group and *N. perurans*) protein databases in total (Figure 1). A total of 33 proteins were found to be significantly differentially expressed *(p* < 0.05) between the exoproteomes of the attenuated and virulent parasites. A total of 11 *N. perurans* (Table 1) proteins and putative homologous protein associated with *Planoprotostelium fungivorum* contributed to the differential proteomic expression observed between parasite exoproteomes and the remaining 22 protein identifications were assigned to bacteria, from genera *Pseudoaltermonas* spp. and *Vibrio* spp. All bacteria-associated protein identifications, significant and non-significant, are listed in Appendix A. Table 2 depicts the top seven bacterial protein identifications, based on relevance to extracellular function and cytotoxicity potential to RTgill W1.

Only one protein, flagellin of *Vibrio tasmaniensis* (strain LGP32), displayed significant upregulation in the virulent exoproteome, meaning 32 proteins remained significantly downregulated in the virulent exoproteome. Thus indicating a diminished role for secreted proteins in *N. perurans* virulence repertoire. Contrarily, the attenuated exoproteome maintained upregulation in these 32 proteins, including all 11 *N. perurans* protein identifications. Of the 33 significant proteins, three *N. perurans* proteins were identified as hypothetical proteins and two *N. perurans* were assigned ‘NA’ or unidentified. Five non-significant proteins that were shared between parasite exoproteomes were maltoporin, PPM-type phosphatase domain-containing protein, ferrichrome-iron receptor, flagellin, and putative ferric enterobactin receptor. A PPM-type phosphatase domain-containing protein (*P. fungivorum*) was the only non-*N. perurans* amoebae protein identified. This identification is likely to represent a protein that has high homology to a native *N. perurans* phosphatase and was, therefore, grouped with the *N. perurans* proteins in Table 1. Proteins that were upregulated in the attenuated exoproteome relate to serine protease and endopeptidase activity, cytoskeletal and membrane associated proteins and suggest functional roles in nutritional acquisition.

### 3.2. Enzymatic Characterisation and Inhibition Assay

All exoproteome isolates hydrolysed the non-specific protease substrate, azocasein, relative to the positive control (Figure 2). The attenuated and virulent exoproteome exhibited different protease activity, with the attenuated having significantly more protease activity than that of the virulent (*p* < 0.0001) exoproteome.

Presence of inhibitors in both the attenuated (*p* < 0.0001) and virulent (*p* < 0.0284) exoproteome significantly reduced protease activity, except for EDTA in the virulent exoproteome, which was found to be non-significant. The introduction of PMSF to both the exoproteome samples resulted in protease inhibition, which suggests the presence of serine and threonine proteases *N. perurans* exoproteome.

### 3.3. Cytotoxicity Assay

The attenuated exoproteome caused significant cell cytotoxicity compared with the virulent exoproteome, revealed by the metabolic impairment dye, Alamar blue (Figure 3). The RTgill W1 cells treated with the virulent exoproteome load remained largely unaffected by the overnight incubation. This result indicates that the virulent isolate of *N. perurans* does not release cytotoxic molecules when cultured in the laboratory with MYA agar.

## 4. Discussion

The aim of this study was to characterise the differences in protein expression in the exoproteome of an attenuated versus a virulent culture of *N. perurans*. Biochemical tests using a protease assay and protease inhibitors were performed to detect and identify specific protease families in all exoproteomes. Previous studies of pathogenic amoebae, such as *E. histolytica* [14] and *N. fowleri* [35], report the cytotoxic role of the secretions from the respective parasites during infection, therefore, this study also sought to investigate the cytotoxicity of *N. perurans* exoproteomes using the epithelial cell line, RTgill W1.

Encompassing the complexity of *N. perurans* and its microbiota, protein databases of several key genera were included from a recent *N. perurans* microbiome study utilising these two amoeba isolates [20]. Furthermore, to facilitate the identification of *N. perurans* proteins the Amoebozoa reference proteome from UniProt was included in the analysis, in conjunction with the *N. perurans* database (20,887 proteins (v2, 07/08/2019, CSIRO) [20]). Overall, the LC-MS/MS results revealed a stark contrast between exoproteomes of attenuated and virulent cultures and most remarkably, that the attenuated culture significantly secretes more proteins than the virulent. A disproportionate quantity of proteins identified originated from commensal bacteria associated with *N. perurans* that are actively releasing proteins into their surrounds, intra- and extracellular to the amoeba. The significant presence of bacterial protein identifications (11 *N. perurans* proteins to 22 bacterial proteins) highlights the intrinsic yet predominant role of the parasite’s microbiome in the attenuated culture and has clear implications when performing comparative exoproteomic analysis. Only one protein, flagellin of *Vibrio tasmaniensis* (strain LGP32), a non-amoebae protein, was found to be present at a significantly higher level in the virulent exoproteome. Flagellin is the main subunit of the bacterial flagellae and is likely present due to the significant presence of *Vibrio* bacteria in the culture. Although present at non-significant elevated levels, five additional proteins were found in both exoproteomes, PPM type phosphatase and four membrane-associated proteins. These membrane-associated proteins; maltoporin, ferrichrome-iron receptor, flagellin, and putative ferric enterobactin receptor were associated with the *Vibrio* and *Pseudoaltermonas* genera. Interestingly, both maltoporin and ferrichrome-iron receptor were found exclusively in the virulent exoproteome. Given their function, these proteins could potentially be bacterial-associated virulence factors of *N. perurans*. Maltoporin is a conserved antigen in *Vibrio* species [36] and ferrichrome-iron receptor is a known bacterial virulence factor associated with iron-scavenging in environments with low iron-bioavailability [37]. The presence of iron acquisition proteins reflects the iron-limited growth conditions in which both *N. perurans* cultures are cultivated in, which suggests that the parasite may rely on its microbiota to scavenge iron.

The majority of protein identifications which were present at a significantly higher level, came from the attenuated exoproteome, and mostly were of bacterial association. However, *N. perurans* specific proteins were identified, many of which had endopeptidase activity which correlates with cytotoxicity, such as Cn-Zn superoxide dismutase (SOD) and physarolisin, (Table 1). SOD expression has previously been found to be significantly higher in the cytoplasmic fraction of a wild type isolate of *N. perurans* [20]. Although SOD is reported to defend parasites against the host response, its presence here may be part of the attenuated isolate’s defence mechanism in response to excess production of endogenous oxidative stressors from its own commensal microbiota.

TolA-like and urea ABC transporter proteins were present at significantly higher levels in the attenuated exoproteome, highlighting that membrane-associated proteins can be released into the extracellular milieu. Presence of ABC transporters have been associated with the exoproteomes of marine bacteria in nutrient poor environments, where increased transcription of transporter proteins increases scavenging potential for nutrients [38,39]. TolA-like protein in *Acanthamoeba* has been described as an environmental adaptation to bacterial toxins [40], a role that *N. perurans* may require during prolonged culture and, thus, explains the increased presence in the attenuated exoproteome.

Physarolisin, a carboxyl enzyme isolated belonging to the serine peptidase family S53 [41] was elevated in the attenuated exoproteome. A cysteine peptidase family (C53) was notably elevated in a newly acquired isolate from its cytoplasmic proteome as a zymogen, discussed in Ní Dhufaigh et al. [20]. Although this cysteine protease was not identified in this current study, the potential for identification of different virulence phenotypes based on differential expression of protease families could be a useful biomarker of virulence. A peptidase, S8 and S53 domain containing protein was absent entirely from the virulent exoproteome but remained elevated in the attenuated exoproteome. However, a peptidase S8/S53 subtilisin kexin sedolisin from *Pseudoaltermonas citrea* was present in both exoproteomes and this protein may explain the serine protease activity seen in both exoproteomes. Multifunctional chaperone expression relating to proteomic homeostasis [42] was increased in the attenuated exoproteome, which may be a result of maintaining protein stability from the commensal microbiotas nutritional stressors. However, most of the differentially expressed proteins of the virulent and attenuated exoproteome were of bacterial origin. The significantly higher level observed in the attenuated *N. perurans*’ exoproteome can be considered an artefact of prolonged culture and a result of the parasite’s associated microbiota secreting soluble components into the culture supernatant. Both attenuated and virulent isolates were xenic cultures and the presence of commensal bacteria is evident in the cultures of attenuated isolates where biofilm growth was routinely observed.

Based on these results, it would appear that the virulent *N. perurans* isolate analysed here, does not secrete an abundance of proteins into its environment. This contrasts with observations made by Butler and Nowak [18], and Bridle et al. [19] that indicated the presence of an extracellular product was inducing a cytopathic effect on the RGE-2 and CHSE-214, respectively. We have previously shown that *N. perurans* cultured for 70 days on MYA can induce AGD [20]. This suggests that the virulent *N. perurans* isolate analysed in this study before harvesting and isolation of its extracellular proteins (ECP) should have exhibited virulent cytopathology. These contrasting results indicate that either, secreted proteins do not play a role in inducing AGD in vivo or, more likely, that the absence of a host signal from salmon gills, mucus or cell (in vitro) signals do not favour the secretion of virulence factors. The parasite may require the presence of certain environmental stimuli to activate such a mechanism. This has been shown in other studies, e.g., Goncalves et al. [15] and Rubin et al. [23] who reported that cultivation of parasites in vitro can facilitate or lessen protease secretion either by providing optimal nutritional or sustained growth. It has also been shown that parasitic amoebae can be prompted to release cytotoxic proteases in response to sugars or host derived components. Mannose added to cultures of *Acanthamoeba* spp. triggers the release of MIP133, a 133 kDa serine protease [43] and, in response, to co-incubation with extracellular matrix components of host fibronectin, *E. histolytica* adheres to these fibronectin fragments, and simultaneously causes proteolysis [44]. Future endeavours relating to the study of the *N. perurans* ECP should explore culture media supplementation. Optimising the conditions required for secretion of virulence factors may reduce the complexity of the ECP by reducing bacterial growth and, furthermore, reduce the presence of secreted bacterial proteins in the proteomic analysis.

Proteases are well described virulence factors of pathogenic amoeba [14] hence, this study aimed to confirm the presence and activity of putative *N. perurans* extracellular proteases by the azocasein assay. Protease activity remained significantly higher (*p* < 0.0001) in the attenuated exoproteome compared with the virulent exoproteome. Based on the protein identifications from LC-MS/MS, we propose the involvement of physarolisin, peptidase S8, and S53 domain-containing protein (of *N. perurans*) and peptidase S8/S53 subtilisin kexin sedolisin (of *P. citrea*) from the attenuated culture to the observed protease activity. Protease activities of both exoproteomes were significantly inhibited in the presence of PMSF indicating serine proteases are dominant in the *N. perurans* exoproteome. The unknown protein, PPER_00011993, was identified in both exoproteomes and could plausibly represent an uncharacterised serine type protease, due to the significant inhibition of protease activity in the virulent exoproteome by PMSF. Peptidase S8/S53 subtilisin kexin sedolisin of *P. citrea* was also present in the virulent culture and may have contributed to the production of protease activity. Protease inhibition using EDTA was also significantly reduced in the attenuated exoproteome however no significant difference in metalloproteinase inhibition was noted in the virulent exoproteome.

RTgill W1 (trout gill) cells served as a model to test the cytotoxic effect of the exoproteomes of both the virulent and attenuated *N. perurans.* Results of the cytotoxicity assays showed that the attenuated exoproteome caused significantly more cytotoxicity than the virulent exoproteome (Figure 3). This is likely to be derived from the presence of a combination of *N. perurans* physarolisin, Cu-Zn superoxide dismutase and bacterial proteins: Peroxiredoxin 2 (of *P. haloplanktis*) secreted alkaline phosphatase (of *P. haloplanktis*) and plausibly an immunogenic protein (of *V. tasmaniensis*) in the exoproteome of the attenuated parasite culture. Cells treated with the virulent exoproteome remained unaffected, and levels of cytotoxicity were comparable to untreated controls. In contrast to this finding, an attenuated culture of *N. perurans* was previously reported to cause no cytopathic effect to an embryonic cell line, CHSE-214 [19]. However, discrepancies in findings could be explained by the selection of the cell line model and, furthermore, the xenic method of culturing *N. perurans* prior to the collection of the parasite’s supernatant. Previous studies with bacteria describe a higher susceptibility of CHSE-214 to infection during co-incubation with *Yersinia ruckeri* compared with salmon head-kidney and Atlantic salmon kidney cells [45], as well as fathead minnow and rainbow trout liver cell lines [46]. Menanteau- Ledouble et al. [45] also identified a significant difference between high passage and low passage of CHSE-214, where younger cells adhered more tightly than older cells. Thus, the influence of passage number may have a direct effect on CHSE-214 cells during cytopathic effect studies. RTgill W1 are known precursor stem cells of the gill [34] and under appropriate culturing conditions, secrete mucus from goblet-like cells [47], therefore, representing a more appropriate in vitro model for AGD. Upon commencing this experiment no salmon cell line was commercially available, thus RTgill W1 was selected. However the recent development of two salmon gill epithelial cell lines from Gjessing et al. [48] should facilitate the use of a more AGD intrinsic model in the future.

The xenic method of culturing *N. perurans* prior to the collection of the parasite’s supernatant may influence the parasite’s release of cytotoxic molecules. As described in Section 2.1, both isolates of *N. perurans* were cultivated on MYA as per Crosbie et al. [25], however Bridle et al. [19] maintained *N. perurans* in L-15 media with CHSE-214 cells for 24 h prior to the collection of the parasite’s supernatant. The change in culture media could have induced a secretion triggering mechanism from a more nutrient rich source than the standard culturing protocol according to Crosbie et al. [25]. To validate this hypothesis, proteomic work should be undertaken on an *N. perurans* exoproteome in L-15 medium culture to fully validate the influence of culture media on exoproteome secretion and hence virulence factor production. Based on the collective results surrounding the attenuated exoproteome, we suggest the microbiome of the attenuated isolate sustains immunogenicity in culture and is responsible for the observed cytotoxicity here but is incapable of establishing AGD in vivo. These findings have direct implications for future in vitro work using *N. perurans* isolates of different age and presumably sampling location.

## 5. Conclusions

In summary, the *N. perurans* exoproteome differs in protein quantity, protease activity, and cytotoxicity between the phenotypes of an attenuated and virulent isolate used in this study that originated from the west coast of Ireland. Presently, the collective results suggest the exoproteome of *N. perurans* is not discriminative of virulence due to the complexity surrounding the comparative attenuated model and its associated microbiota. In essence, this study represents an evaluation of a long-term cultured isolate of *N. perurans*. A total of 33 proteins were found to be significantly differentially expressed *(p* < 0.05) between the exoproteomes of the attenuated and virulent parasites, with 32 of these proteins downregulated in the virulent parasite. Proteins that were non-significantly shared in both exoproteomes were maltoporin, ferrichrome-iron receptor, and putative ferric enterobactin receptor. The significant inhibition of protease activity in both exoproteomes in the presence of PMSF suggests serine proteases are dominant in the *N. perurans* exoproteome. As previously stated, we cannot discard the possibility that other virulent *N. perurans* isolates in enriched culture media, as was the experimental design in Bridle et al. [19], would reveal the extracellular product previously hypothesised. Furthermore, *N. perurans* may require a contact dependent mechanism by adhering to the host in order to trigger the extracellular product. Going forward with this work, we suggest exploring additional means of *N. perurans* culture supplementation and mechanisms to reduce the bacterial secretion effect observed in the attenuated culture.

## Figures and Tables

**Figure 1 microorganisms-09-01258-f001:**
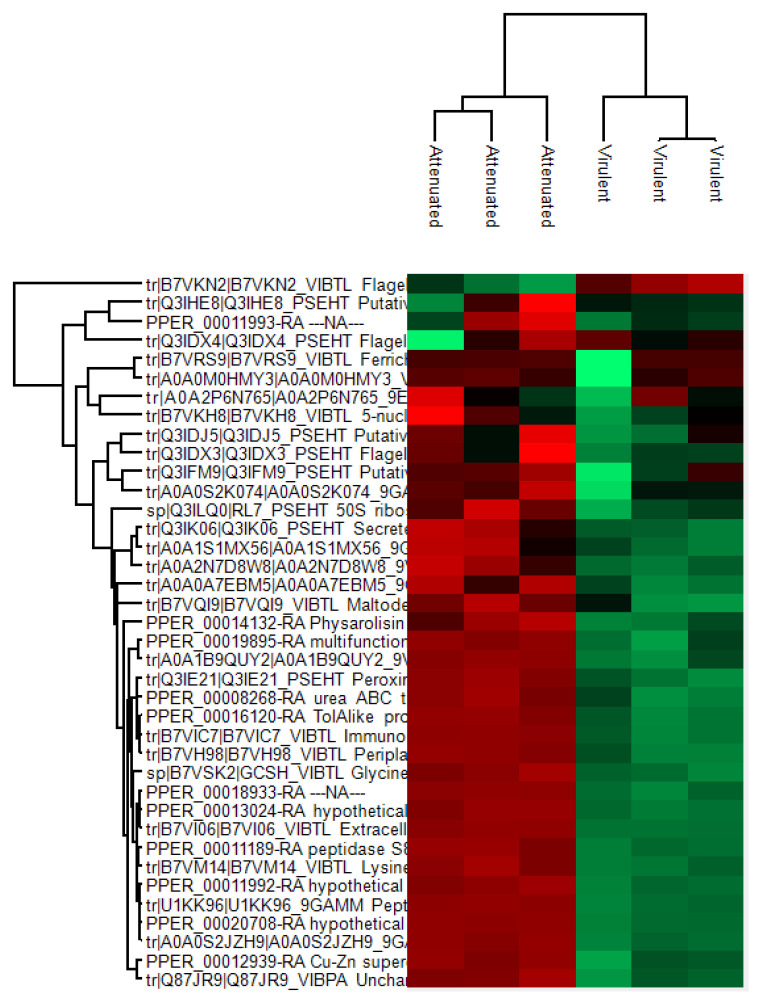
Hierarchical clustering of the 38 protein identifications between the attenuated and virulent *Neoparamoeba perurans* exoproteome replicate samples. Label free quantification (LFQ) intensity values of the 33 significant proteins were normalised by z-score with red colours showing increased abundance and green displaying a decreased expression of proteins. A detailed list of proteins clustered here can be found in Appendix A.

**Figure 2 microorganisms-09-01258-f002:**
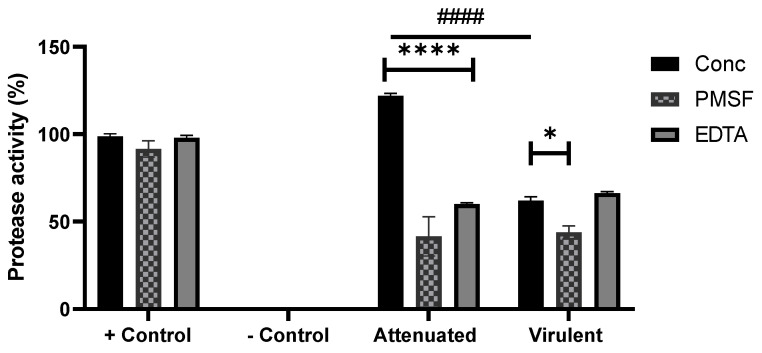
Protease activity (conc) and inhibition assay (PMSF, EDTA) of *Neoparamoeba perurans* exoproteome samples from the attenuated and virulent cultures using azocasein (non-specific protease substrate) as a substrate. Trypsin (2 mg/mL) was used as positive control and sterile seawater as a negative. Results are expressed as a percent of the positive control. All datasets are represented as ±SEM from biological replicates. Two-way ANOVA followed by a Tukey post hoc test for multiple comparisons: Attenuated vs. phenylmethylsulfonyl fluoride (PMSF) and ethylenediaminetetraacetic acid (EDTA), **** (*p* < 0.0001); Virulent vs PMSF, * (*p* < 0.0284), Attenuated vs. Virulent, #### (*p* < 0.0001).

**Figure 3 microorganisms-09-01258-f003:**
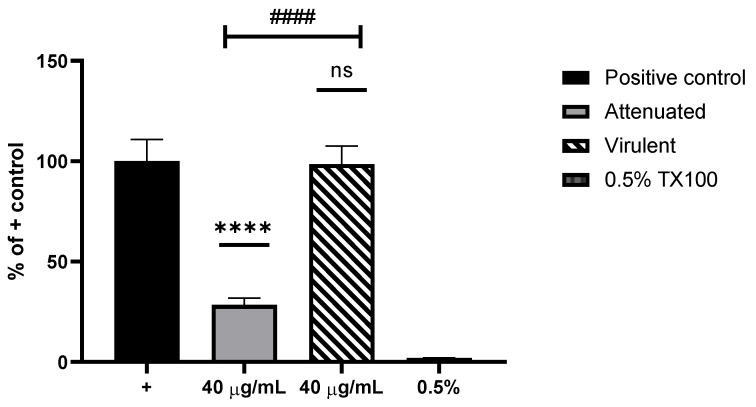
Cytotoxicity assay using RTgill W1 and 40 µg/mL of the extracted *Neoparamoeba perurans* attenuated and virulent exoproteomes. RTgill W1 cells in L-15 were used as positive (+) control and 0.5% Triton X-100as a negative. Results are expressed as a percent of the positive control. All data sets are represented as ±SEM from biological replicates. One-way ANOVA followed by Tukey post hoc test for multiple comparisons: Attenuated vs + control, **** (*p* < 0.0001); Virulent vs + control, ns; Virulent vs +control, ns; Attenuated vs virulent, #### (*p* < 0.0001).

**Table 1 microorganisms-09-01258-t001:** *Neoparamoeba perurans* protein identifications and a putative homologous protein associated with *Planoprotostelium fungivorum* identified in attenuated and virulent exoproteomes, characterised by a two-way Students *t*-test in Perseus.

Gene Bank IDs	Protein IDs	Fasta Headers	Mol. Weight (kDa)	Fold Change ^1^	Atten Peptides ^2^	Vir Peptides ^2^	*t*-Test ^3^
PRP79790.1	A0A2P6N765	PPM-type phosphatase domain-containing protein (*Planoprotostelium fungivorum*)	190.44	−0.144883	14	19	NS
MW534129	PPER_00018933-RA	NA	281.31	−4.72831	40	17	+
MW534130	PPER_00016120-RA	TolAlike protein	36.42	−3.89208	11	0	+
MW534131	PPER_00020708-RA	Hypothetical protein	64.3	−5.01281	10	0	+
MW534132	PPER_00011993-RA	NA	35.25	−1.15085	8	6	+
MW534133	PPER_00013024-RA	Hypothetical protein	13.01	−4.76979	8	3	+
MW534135	PPER_00019895-RA	Multifunctional chaperone	28.08	−3.49453	9	2	+
MW534136	PPER_00008268-RA	Urea ABC transporter substrate-binding protein Branched-chain amino acid	47.26	−3.18528	6	0	+
MW534137	PPER_00011992-RA	Hypothetical protein	39.76	−2.86775	6	6	+
MT419758	PPER_00012939-RA	Cu-Zn superoxide dismutase	15.50	−3.72917	6	2	+
MW534138	PPER_00014132-RA	Physarolisin	49.49	−4.06437	5	1	+
MW534139	PPER_00011189-RA	Peptidase S8 and S53 domain-containing protein	58.69	−2.78376	3	0	+

The + sign = statistically significant differential proteins (based on q-value); NS = non-significant; ^1^ Fold change of virulent proteins related to attenuated proteins; ^2^ Razor and unique peptide counts (attenuated and virulent); ^3^ Student’s *t*-test significance.

**Table 2 microorganisms-09-01258-t002:** Non-*Neoparamoeba perurans* proteins of bacterial origin identified in attenuated and virulent *N. perurans* exoproteomes. Proteins are ranked based on relevance to extracellular function. The complete set of identifications are provided in Appendix A.

Gene	Protein IDs	Fasta Headers	Organism	Mol. Weight (kDa)	Fold Change ^1^	Att. Peptides ^2^	Vir Peptides ^2^	*t*-Test ^3^
ST37_14230	A0A1X1MR06	ABC transporter substrate-binding protein	Vibrio sp. qd031	47.26	−3.18528	6	0	+
VS_II0220	B7VQI9	Maltodextrin-binding protein	Vibrio tasmaniensis (strain LGP32	42.39	−4.90439	13	4	+
VS_0355	B7VIC7	Immunogenic protein	Vibrio tasmaniensis (strain LGP32)	34.44	−5.20998	11	1	+
ahpCB	Q3IE21	Peroxiredoxin 2 (TSA) (PRP)	Pseudoalteromonas haloplanktis (strain TAC 125)	22.12	−4.75659	9	0	+
PSHAa297; B1199_19990	Q3IK06; A0A244CKW5	Secreted alkaline phosphatase; Alkaline phosphatase	Pseudoalteromonas haloplanktis (strain TAC 125); Pseudoalteromonas ulvae	56.1	−4.1791	10	2	+
PCIT_22080	U1KK96	Peptidase S8/S53 subtilisin kexin sedolisin	Pseudoalteromonas citrea DSM 8771	127.37	−3.70559	6	3	+

The + sign = statistically significant differential proteins (based on q-value); ^1^ Fold change of virulent proteins related to attenuated proteins; ^2^ Razor and unique peptide counts (attenuated and virulent); ^3^ Student’s *t* test significance.

## Data Availability

All data supporting this study are included in the results section of the manuscript and the Appendix A and are openly available in public databases. The mass spectrometry proteomics data have been deposited to the ProteomeXchange Consortium via the PRIDE partner repository [49] with the dataset identifier PXD025154.

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
