# Peer review of "Differential Exoproteome and Biochemical Characterisation of Neoparamoeba perurans"

_microorganisms, 2021, doi:10.3390/microorganisms9061258_

Round 1
Reviewer 1 Report
The paper by Dillon et al on the exoproteome (extracellular products) of the aetiological agent of AGD, Neoparamoeba perurans, is a useful manuscript presenting data that will be of much interest to scientists in the field of AGD. The paper highlights the complexity of determining amoebae parasitic virulence factors, even after establishing an attenuated phenotype in culture. I commend the authors for their approach to understand the parasite's mechanisms of virulence and annotate associated biomarkers.
The manuscript is well structured and well written. The science is also sound, but my main concerns relates to whether the 'virulent' strain has actually be determined as virulent. I'm not sure whether the authors have established this by challenging fish with this strain vs. the long-term cultured strain. It would be useful for the authors to include more information about the condition of the fish from the which the 'virulent strain' was collected. Even some information on AGD score would be useful, and duration of the challenge and the challenge strain used (unless of course the fish were naturally infected).
The condition of the amoebae should also be described. Can the authors rule out that the cytotoxicity caused by the attenuated strain is not related to established microbial communities in the long-term cultured strain?
Other than these scientifically orientated questions I have no major concerns with the manuscript and some clarity on these issues and perhaps additional discussion would suffice to make the manuscript suitable for publication.
Specific points:
Lines 72-73 - Please include more details on the diseased fish. What AGD score were the fish exhibiting (i.e. in accordance with Taylor et al., 2009)? How long were the fish infected? Were there any other disease issues in these fish?
Lines 80-83 - What was the condition of the isolated amoebae? Please include any crude microscopic observations. For example were amoebae active - presence of plasmodia, feeding etc.
Section 2.1 - How can the strain be defined as 'virulent'? Has virulence been established by challenging fish after the 70 days of culture?
Lines 167-169 - Why were other protease families not investigated?
Section 2.9 - line 196 - typo on Preparation
Figure 1 - Not clear annotation. I suggest shortening names/or abbreviating then including a Table with full annotations (or legend) including accession numbers.
Figure 2 - Does PMSF not also inhibit cysteine protease activity? Also line 384
Figure 3 - Can you rule out cytotoxicity not being caused by bacteria associated with long-term microbial colonisation associated with 'attenuated' strain?
Further clarity and/or discussion on these points would be really useful!
Author Response
Many thanks for reviewing our publication. Addressing your concern regarding the virulent isolate, this isolate was collected from naïve fish (the text now reflects this) and was cultured for 70 days. I have included in the text that this virulent isolate was proven as virulent in a separate challenge trial held in Galway-Mayo Institute of Technology therefore we can confirm its virulence. The attenuated isolate, as mentioned was confirmed as having lost virulence in our previous publication, Ní Dhufaigh et al., (2021) during a challenge trial. This paper is cited within the text so readers can access the paper for further information regarding the challenge trial and gill scoring.
I mention in lines 401-407:
“Results of the cytotoxicity assays showed that the attenuated exoproteome caused significantly more cytotoxicity than the virulent exoproteome (Figure 3). This is likely to be derived from the presence of a combination of N. perurans physarolisin, Cu-Zn superoxide dismutase and bacterial proteins: Peroxiredoxin 2 (of P. haloplanktis) secreted alkaline phosphatase (of P. haloplanktis) and plausibly an immunogenic protein (of V. tasmaniensis) in the exoproteome of the attenuated parasite culture”.
I am therefore suggesting there is a cytotoxic role of the bacterial species within the attenuated microbiome, due the identification peroxiredoxin 2 of P.haloplanktis and an alkaline phosphatase of V. tasmaniensis.
Addressing specific points, lines 72-73, I have included that these were naïve Atlantic salmon smolts that were affected by AGD. This was confirmed with gill scoring by a vet and a fish farm manager. The fish did not display any other signs of disease. This method of isolating amoeba is the common practice in Ireland. After culturing of N. perurans has passed for 1-2 weeks we validated that we had the correct amoeba species in house using qPCR (Downes et al., 2015). Route culturing also involved inspecting amoeba using confocal microscopy, adding visual confirmation by observing the characteristic pseudopodia shape of N. perurans. For lines 80-83, I have added in
“Amoebae exhibited characteristic pseudopodia and were sub-cultured weekly by transferring free-floating cells to fresh MYA plates”
This now appears on lines 86-87. N. perurans attaches and lifts into suspension during culturing which is how the community sub-cultures the amoeba without the need for trypsin-EDTA.
In section 2.1, I have explained in the methods that the virulent isolate was validated as virulent during a separate challenge trial to this work.
For lines 167-169, we did not expand the scope beyond serine and metalloproteinases as we reviewed the publication of Khan et al., (2000), that described protease activity inhibition by PMSF of the secretion fraction of Acanthamoeba, a free living opportunistic protest, via gel zymography. In future work, it would be beneficial to fully assess all protease families.
For Figure 1, the protein headers do not appear as legible, this is common with heat maps that are generated in Perseus software. I have updated the legend with the full list of proteins are included in Table 4 of the supplementary material.
For Figure 2 -, PMSF is well described as a serine protease inhibitor but it can also inhibit cysteine proteases with serine residues. Papain and mammalian acetylcholinesterase can be inhibited with PMSF and are cysteine proteases however, most researchers would avail of a cysteine inhibitor such as E64 for broad acting cysteine protease inhibition. I can add that PMSF will inhibit cysteine proteases with serine residues but I believe its incorrect to state all cysteine proteases would be inhibited with PMSF. I refer you to Khan et al., (2000) paper showing that the authors here also only referred to serine protease inhibition:
“In some experiments, samples were pretreated with phenylmethylsulfonyl fluoride (PMSF, an inhibitor of serine protease; 1 mM), aprotonin (a serine protease inhibitor; 5 U/ml)”.
For Figure 3 – We cannot rule out the possibly that the bacteria within the attenuated is causing the cytotoxicity. I refer back to the protein identifications wherein bacterial proteins of putative cytotoxicity were identified :
“Results of the cytotoxicity assays showed that the attenuated exoproteome caused significantly more cytotoxicity than the virulent exoproteome (Figure 3). This is likely to be derived from the presence of a combination of N. perurans physarolisin, Cu-Zn superoxide dismutase and bacterial proteins: Peroxiredoxin 2 (of P. haloplanktis) secreted alkaline phosphatase (of P. haloplanktis) and plausibly an immunogenic protein (of V. tasmaniensis) in the exoproteome of the attenuated parasite culture”
References
Dhufaigh, K. N., Dillon, E., Botwright, N., Talbot, A., O’Connor, I., MacCarthy, E., & Slattery, O. (2021). Comparative proteomic profiling of newly acquired, virulent and attenuated Neoparamoeba perurans proteins associated with amoebic gill disease. Scientific reports, 11(1), 1-12.
Downes, J. K., Henshilwood, K., Collins, E. M., Ryan, A., Connor, I. O., Rodger, H. D., ... & Ruane, N. M. (2015). A longitudinal study of amoebic gill disease on a marine Atlantic salmon farm utilising a real-time PCR assay for the detection of Neoparamoeba perurans. Aquaculture Environment Interactions, 7(3), 239-251.
Khan, N. A., Jarroll, E. L., Panjwani, N., Cao, Z., & Paget, T. A. (2000). Proteases as Markers for Differentiation of Pathogenic and Nonpathogenic Species ofAcanthamoeba. Journal of Clinical Microbiology, 38(8), 2858-2861.
Reviewer 2 Report
Please see attached report

Author Response
Reviewer Two.
Many thanks for reviewing our publication. Please see our response to your concerns:
For Ln. 36: The statement regarding ‘phylogenetically inseparable’ was a direct quote from the publication describing this research (Feehan et al., 2013),
“Morphology and closest phylogenetic affinities suggest that P. invadens would be assignable to the genus Neoparamoeba; however, nuclear SSU rDNA trees show that Neoparamoeba and Paramoeba are phylogenetically inseparable”. We can change this if you wish but we paraphrased from the study this work originated in.
For Lns. 51-55- I have moved the study design to the end of the introduction paragraph.
For . Ln. 125 – delete “were included”- has been deleted.
For 4. Lns. 180-181 – The cells were cultivated in 100% atmospheric O2 with no CO2 inputted. The incubator that was used was a water jacketed CO2 Incubator. The full description of the incubator was provided for reader transparency but I’ve deleted this information regarding the CO2 incubator as I can see how this is confusing.
For Ln. 185 - cell number enumerated has been changed to cell number counted.
For Ln. 187 -Fluorescent 96-well plate has been updated to ‘a 96-well fluorescence plate’
For Ln 199 “assay performed three times independently” – this relates to three independent technical replicates that included three biological replicates on each plate .
For. Lns. 208-209. The heatmap has been adjusted to state the 38 proteins identified, not just significant. I have also included “A detailed list of proteins clustered here can be found in Table 4 of the Supplementary material”, as to direct readers so they can see the full protein headers.
Lns 209-210: This now reads as “A total of 11 N. perurans (Table 1) proteins and putative homologous protein associated with Planoprotostelium fungivorum” to match the figure legend.
For. Ln 253 (Fig. 2) The concentrated and inbitors have been added to the figure legend “Figure 2. Protease activity (conc) and inhibition assay (PMSF, EDTA) of Neoparamoeba perurans exoproteome samples from the attenuated and virulent cultures using azocasein (non-specific protease substrate) as a substrate”
For point 11 . The difference between the positive control and the attenuated may be due to two possibilities ; 1) One possible reason to account for the dramatic difference despite the vast difference in concentration could relate to increased specific activity (enzyme U/mg of protein) for the enzymes in the LA sample compared with the bovine trypsin. Similar enzymes from different organisms can have different levels of activity. Our positive control was trypsin, it was at a high concentration so ideally it should have had the highest activity without question. However, it is possible, given the range of organism’s present, that one or more have secreted proteases that have greater specific activity than the trypsin. The collective impact may push the enzyme activity over that of trypsin despite the difference in concentration. 2) the substrate in question is a non-specific substrate, meaning numerous proteases can hydrolyse the casein dye therefore the attenuated exoproteome may have more than serine rich proteases and may contain, for example cysteine proteases. We have added on line 276 that azocasein is a non-specific dye.
For Lns 263-265. The sentence has been updated to “The introduction of PMSF to both the exoproteome samples resulted in protease inhibition, which suggests the presence of serine and threonine proteases N. perurans exoproteome”.
- Lanes 266-279. For Fig. 3: My sincere apologies, this is the incorrect graph for the cytotoxicity assay. Please allow me to explain. I also have a set of results that I did not include in this paper and this is the graph belonging to those results. The graph you have reviewed was of antibiotic treated cultures, IE the attenuated and virulent were subcultured on MYA that was supplemented with penicillin-streptomycin. This was done in triplicate but the proteomic data to match this work was only done once. Therefore I had no means of performing any statistical analysis on the protein dataset so I did not want to include the cytotoxicity ‘antibiotic treated’ assay without the matching protein results. The correct graph is now in place and this is what the manuscript text describes ie that the attenuated exoproteome is more cytotoxic. My sincere apologies again, I am not sure how such a mistake was made.
Point 14. The positive control was L-15 and the Alamar blue solution was made in sterile PBS so all media was removed and the 5% solution of Alamar blue made in PBS was added to each well. I was refereeing to the cytotoxicity assay in which the wells received the alamar blue solution in PBS in the legend of figure 3 but I can see that this is confusing. I have updated the legend to say “RTgill W1 cells in L-15 were used as positive (+) control”. Figure 2 legend is not transposed here in figure 3.
Point 15. Lns. 308-315: The mention of maltoporin and ferrichrome-iron recptor was for both cultures. The text has been updated to include both cultures: “The presence of iron acquisition proteins reflects the iron‑limited growth conditions in which both N. perurans cultures are cultivated in, which suggests that the parasite may rely on its microbiota to scavenge iron”.
Point 16. Ln 315, the typo “in in vitro” has been removed, now reads as “in vitro”
- Ln 366: The typo - “either by providing optimal nutritional growth or sustained growth” has been corrected to “lessen protease secretion either by providing optimal nutritional or sustained growth”.
- Lanes 393-395. And Point 19 , Lns 399-400. The correct graph is in place and the manuscript text reflects the results.
References:
Feehan, C. J., Johnson-Mackinnon, J., Scheibling, R. E., Lauzon-Guay, J. S., & Simpson, A. G. (2013). Validating the identity of Paramoeba invadens, the causative agent of recurrent mass mortality of sea urchins in Nova Scotia, Canada. Diseases of aquatic organisms, 103(3), 209-227.
Round 2
Reviewer 2 Report
Please see comments in the attached document.

Author Response
Reviewer Two.
Please note that this revised version has addressed most of the points raised in my 1st review. There are only a few points pending correction/clarification. For my convenience, I am using the author’s answer to my original review but I have taken the liberty to erase minor points, typos etc. already corrected.
For Ln. 36: The statement regarding ‘phylogenetically inseparable’ was a direct quote from the publication describing this research (Feehan et al., 2013), […] “We can change this if you wish but we paraphrased from the study this work originated in.
[Reviewer] Understood, it is fine as a direct quote.
Kerrie : Ok, thank you.
For Lns. 51-55- I have moved the study design to the end of the introduction paragraph.
[Reviewer] Please note that after this change the order of the citations [18-end] needs to be rearranged in the whole manuscript. I noticed that in the revised version you have done it after introducing the new reference [25] to account for other referee’s suggestion, but I am afraid that it needs to be done again.
Kerrie: I have corrected this, thank you for spotting this issue.
For. Lns. 208-209. The heatmap has been adjusted to state the 38 proteins identified, not just significant. I have also included “A detailed list of proteins clustered here can be found in Table 4 of the Supplementary material”, as to direct readers so they can see the full protein headers.
[Reviewer] This is clear now.
Kerrie: ok, thank you.
Lns 209-210: This now reads as “A total of 11 N. perurans (Table 1) proteins and putative homologous protein associated with Planoprotostelium fungivorum” to match the figure legend.
[Reviewer] This still does not clarify the fact that in table 1 there are 12 (not 11) N.perurans proteins and putative homologous associated to P.fungivorum. At least attending to the GenBank IDs and Protein IDs. Please note that two different GenBank ID (MW534133 and MW534134) and Protein IDs (PPER_00013024-RA and PPER_00016390) appear to share the same row in the 1st three columns, but there are actually 2 values separated by a semicolon. Is this a table rendering problem? Are these protein IDs synonymous? Please correct as appropriate and fix the number of proteins stated in the text (also in other sections of the MS if appropriate) and the rendering of the table.
Kerrie: Yes you are correct, a double entry was mistakenly added to this table. This is removed from the table.
For point 11 . The difference between the positive control and the attenuated may be due to two possibilities ; 1) One possible reason to account for the dramatic difference despite the vast difference in concentration could relate to increased specific activity (enzyme U/mg of protein) for the enzymes in the LA sample compared with the bovine trypsin. Similar enzymes from different organisms can have different levels of activity. Our positive control was trypsin, it was at a high concentration so ideally it should have had the highest activity without question. However, it is possible, given the range of organism’s present, that one or more have secreted proteases that have greater specific activity than the trypsin. The collective impact may push the enzyme activity over that of trypsin despite the difference in concentration. 2) the substrate in question is a non-specific substrate, meaning numerous proteases can hydrolyse the casein dye therefore the attenuated exoproteome may have more than serine rich proteases and may contain, for example cysteine proteases. We have added on line 276 that azocasein is a non-specific dye.
[Reviewer] This still does not clarify why the inhibition assay of the positive control using PMSF hassuch a minor effect. From Fig. 2 it looks like 1mM PMSF only reduces Trypsin activity by 5-10%! I can understand that having a complex mixture of proteases in the exoproteome samples against a nonspecific substrate can result in higher protease activity than the positive control, but what worries me is the failure to inhibit the trypsin by PMSF. I am not an expert in these assays so I’ll just take the author’s word that this does not lead to question the QC of the assay.
Kerrie: I allowed for only a 2 minute PMSF incubation with each sample, trypsin control and PBS control. I have since read a minimum time of 30 minutes may be required to fully inactivate the activity of trypsin in azocasein. It was however, effective for the experimental samples.